# The Anti-Fibrotic Effect of Cold Atmospheric Plasma on Localized Scleroderma In Vitro and In Vivo

**DOI:** 10.3390/biomedicines9111545

**Published:** 2021-10-26

**Authors:** Stephanie Arndt, Petra Unger, Anja-Katrin Bosserhoff, Mark Berneburg, Sigrid Karrer

**Affiliations:** 1Department of Dermatology, University Hospital Regensburg, Franz-Josef-Strauss Allee 11, 93053 Regensburg, Germany; petra.unger@ukr.de (P.U.); mark.berneburg@ukr.de (M.B.); sigrid.karrer@ukr.de (S.K.); 2Institute of Biochemistry, University of Erlangen-Nuernberg (FAU), 91054 Erlangen, Germany; anja.bosserhoff@fau.de; 3Comprehensive Cancer Center Erlangen-EMN (CCC ER-EMN), 91054 Erlangen, Germany

**Keywords:** cold atmospheric plasma, localized scleroderma, anti-fibrosis, bleomycin-induced fibrosis

## Abstract

Cold Atmospheric Plasma (CAP) has shown promising results in the treatment of various skin diseases. The therapeutic effect of CAP on localized scleroderma (LS), however, has not yet been evaluated. We investigated the effects of CAP on LS by comparing human normal fibroblasts (hNF), human TGF-β-activated fibroblasts (hAF), and human localized scleroderma-derived fibroblasts (hLSF) after direct CAP treatment, co-cultured with plasma-treated human epidermal keratinocytes (hEK) and with an experimental murine model of scleroderma. In hAF and hLSF, 2 min CAP treatment with the MicroPlaSterβ^®^ plasma torch did not affect pro-fibrotic gene expression of alpha smooth muscle actin, fibroblast activating protein, and collagen type I, however, it promoted re-expression of matrix metalloproteinase 1. Functionally, CAP treatment reduced cell migration and stress fiber formation in hAF and hLSF. The relevance of CAP treatment was confirmed in an in vivo model of bleomycin-induced dermal fibrosis. In this model, CAP-treated mice showed significantly reduced dermal thickness and collagen deposition as well as a decrease in both alpha smooth muscle actin-positive myofibroblasts and CD68-positive macrophages in the affected skin in comparison to untreated fibrotic tissue. In conclusion, this study provides the first evidence for the successful use of CAP for treating LS and may be the basis for clinical trials including patients with LS.

## 1. Introduction

Localized scleroderma (LS)—also termed morphea—is a rare chronic inflammatory disease of the collagenous connective tissue with an incidence rate ranging from 0.4 to 2.7/100,000/year [1,2]. LS is characterized by skin thickening with increased quantities of collagen in the affected lesion [3]. According to Peterson et al., the LS entity is subdivided into linear, plaque, deep, bullous and generalized morphea [2]. Apart from the skin, neighboring structures such as adipose tissue, muscles, joints and bones can also be affected, depending on the subtype of LS. In contrast to systemic sclerosis (SS), internal organs are not involved.

Collagen balance regulated by dermal fibroblasts plays an important role in LS. The transforming growth factor β (TGF-β) exerts a major influence on the collagen metabolism and has been implicated in research as a crucial factor for the development of fibrosis [4,5]. The release of TGF-β stimulates the production of various extracellular matrix (ECM) components such as collagen type I (Coll I) [6,7]. This process is primarily caused by the inducing effect of TGF-β in combination with mechanical stress and leads to the transdifferentiation of fibroblasts into contractile myofibroblasts [8]. In response to cell damage, myofibroblasts in turn secrete an increased number of signaling molecules, including TGF-β and the alpha smooth muscle actin (αSMA), during the inflammatory phase [9,10]. In addition to the pro-fibrotic property of TGF-β and the associated increased collagen synthesis, the release of growth factors also has a simultaneous inhibitory effect on the production of proteases, including matrix metalloproteinase 1 (MMP-1) [11]. This process prevents enzymatic degradation of ECM and enhances the pro-fibrotic process of excessive collagen production.

Equally important in LS is the function of typical pro-inflammatory cytokines, such as interleukin 1 alpha (IL-1α), interleukin 6 (IL-6), interleukin 8 (IL-8) and the tumor necrosis factor alpha (TNFα). These cytokines are strongly activated in many different cell types, including fibroblasts and keratinocytes [12,13,14]. The synthesis of pro-inflammatory cytokines forms a mutually stimulating network. IL-1α plays a pivotal role in this process because it can influence the collagen metabolism and is a definitive stimulus of the synthesis of other pro-fibrotic regulatory proteins such as IL-6, IL-8 and TNFα [13,15]. Conversely, TNFα also stimulates IL-1α synthesis [12]. Fibroblasts of patients with sclerosis have shown increased levels of IL-1α, which could explain the additional synthesis of other pro-fibrotic cytokines such as IL-6 and the accompanying excessive production of Coll I [15].

So far, no optimal therapy of LS is available because of the rareness, complexity, and heterogeneity of the disease and the fact that its molecular mechanisms have not yet been fully elucidated. In general, early treatment in the active stage (inflammatory phase) of the disease promises a higher chance of success than treatment in the later stages (fibrotic phase) [16]. To determine the appropriate treatment option for each LS patient, European medical specialists have developed a treatment algorithm within the framework of a German and European guideline [17,18].

For patients with limited LS, topical glucocorticoid treatment is recommended, which can be given together with phototherapy [18]; treatment with phototherapy alone is also possible. Ultraviolet A_1_ (UVA_1_), psoralen plus UVA (PUVA) and narrow-band UVB (nbUVB) reach different depths in the skin and are effective in treating limited LS [19,20]. PUVA or UVA_1_ treatment reaches the dermis, but nbUVB can only reach the epidermis. Studies have shown that treatment with UVA_1_ at a medium dose (50 J/cm^2^) is significantly better than treatment with nbUVB [20,21,22]. UVA_1_ phototherapy in particular activates matrix metalloproteinases including MMP-1 in human dermal fibroblasts [23,24,25]. Furthermore, UVA_1_ has a direct inhibitory effect on TGF-β production [26] and consequently also on αSMA [27], which in turn stops the activation of myofibroblasts and subsequent collagen production, thus counteracting fibrosis.

More severe LS subtypes, which additionally affect extracutaneous structures such as generalized, linear, or deep LS, usually require early systemic therapy [20]. Immunosuppressants such as methotrexate (MTX) alone or in combination with systemic corticosteroids have been used successfully [28,29,30,31,32]. In a few studies, LS has been treated with other systemic agents, which are usually used for treating systemic scleroderma (SS), such as penicillamine, mycophenolate mofetil and imatinib [16]. The efficacy and safety of these drugs in this context, however, still need to be verified in large clinical trials. First successful trials on the treatment of LS focused on the multityrosine-kinase inhibitor nintedanib [16] and the Januskinase (Jak)-inhibitor tofacitinib [33,34]. Nintedanib blocks the vascular endothelial growth factor, the platelet-derived growth factor, and the TGFβ-induced proliferation and migration of fibroblasts and decreases the synthesis of ECM proteins [16,35,36]. Tofacitinib blocks IL-4/IL-13, IL-12, IFN-γ and TGF-β activated signals via the Januskinase-signal transducer and activator of transcription (Jak-STAT) pathway, suggesting that the inhibition of these pathways may lead to therapeutic success in this group of diseases [37].

Knowledge about the pathophysiology of SS has rapidly increased over the past years, hence most targeted therapies for SS are still being developed [20]. The question whether these novel targeted therapies can be successfully applied to LS remains unanswered because of the lack of knowledge about the key pathways affected in LS. Two clinical trials using monoclonal antibodies to IL-4Rα (dupilumab) and IL-6R (sarilumab) for the treatment of LS are in progress [20]. Overall, the potential benefit of such novel targeted therapies has to be weighed against possible harmful side effects in non-systemic diseases such as LS. Therefore, effective topical therapies for the treatment of LS remain of great interest.

One such topical therapy is Cold Atmospheric Plasma (CAP). CAP is an ionized gas that is created at room temperature and atmospheric pressure and has complex effects on cells, tissues, and living organisms [38,39,40,41,42,43,44,45]. CAP treatment is contact-free and pain-less. It accelerates wound healing and stimulates skin closure, particularly of chronic wounds [46,47,48,49,50,51,52,53,54]. Nevertheless, the mechanisms of action of CAP on skin components is not yet fully understood. The manner in which the various skin cell subsets are affected and respond to CAP exposure is being investigated in many studies. Effects observed on skin cells are largely associated with the modulation of inflammatory pathways and redox-sensitive pathways due to reactive oxygen and nitrogen species generated by CAP [55,56,57,58].

During wound healing, the relevant cells are fibroblasts, keratinocytes, endothelial cells and immune cells. During tissue injury, CAP treatment affects these cells directly but also transfers signals via autocrine and paracrine cell–cell communication into deeper cell layers by producing and exchanging various wound healing-relevant cytokines (e.g., IL-6, IL-8, TNFα) and growth factors (e.g., TGF-β1/2, VEGF, EGF, FGF) [59,60,61]. This process is followed by the activation of intracellular signaling pathways, thus influencing cellular mechanisms such as migration, proliferation, ECM remodeling, or angiogenesis [59,60,62,63,64,65,66,67].

Interestingly, almost the same cell types, cytokines and growth factors as well as the same cellular mechanisms are involved in provoking the progression of LS. Therefore, we speculated that CAP treatment may also affect the pathophysiology of LS. In general, cytokines and growth factors always strive for a balance between pro- and anti-inflammatory immune responses in their bioactivity. Therefore, anti-inflammatory and anti-fibrogenic effects of CAP would be beneficial for successful LS treatment. However, CAP effects on inflammatory diseases such as LS have not yet been analyzed and are therefore the topic of the present study.

## 2. Materials and Methods

### 2.1. Plasma Device and the Treatment of Cells and Animals

The plasma device used in this study was the MicroPlaSterβ^®^ plasma torch system (Microwave 2.45 GHz, 80 W, argon flow 4.0 L/min, treatment diameter ~5 cm) from Adtec Plasma Technology Co. Ltd., Hiroshima, Japan/London, UK. This device has been successfully used for the treatment of chronic and acute wounds [47,68,69], skin pruritus [42], actinic keratosis [70] in both clinical studies and in different in vitro and in vivo animal models [59,60]. In addition, the evidence that neither toxic nor mutagenic effects on cells could be observed after treatment with the MicroPlaSterβ^®^ device has been published [71]. Tissue or cell damage using the MicroPlaSterβ^®^ or argon gas placebo mode was not reported in any of these studies, which is why we did not perform argon gas controls in this study. A description of the plasma torch, plasma device and optical emission spectrum of the plasma discharge can be found elsewhere [60]. Cells and animals were treated with CAP in analogy to a previous study [60].

### 2.2. Cell Lines and Cell Culture Conditions

Human epidermal keratinocytes (hEK) (#FC-0007; CellSystems^®^ GmbH, Troisdorf, Germany) were cultured in DermaLife K complete kit (#LL-0007; CellSystems^®^ GmbH, Troisdorf, Germany). Human normal fibroblasts (hNF) (#231340; CellSystems^®^ GmbH, Troisdorf, Germany) and fibroblasts isolated from the skin of patients with LS (hLSF) (gift from Rüdiger Hein, Technical University Munich, Munich, Germany) were cultured as described elsewhere [72]. For fibroblast activation (hAF), hNF were treated with 10 ng/mL of human recombinant TGF-β1 (Anprotec, Bruckberg, Germany) in Dulbecco’s modified Eagle’s medium (DMEM; ThermoFisher Scientific, Regensburg, Germany) supplemented with 10% fetal bovine serum (FBS; Anprotec, Bruckberg, Germany) and 1% penicillin/streptomycin (P/S; Sigma-Aldrich GmbH, Steinheim, Germany) as well as 1% L-glutamine (Sigma-Aldrich GmbH, Steinheim, Germany) for 24 h. The activity of hAF was controlled by collagen αI (1) (Coll I) real-time PCR analysis as described in 2.5. Mycoplasma contamination was regularly excluded according to the manufacturer’s instructions of the PCR Mycoplasma Test Kit (PanReac AppliChem, Darmstadt, Germany).

### 2.3. Animals and Treatment Groups

The mouse experiments were conducted with 16 female and 16 male 129Sv/Ev mice (original strain from Robertson Lab of Dunn School Pathology of the University of Oxford, UK). All animals were between 6 and 8 weeks old at the start of the study and were divided into two experimental groups. Group 1 contained 8 female and 8 male mice, which were treated with bleomycin (BLM; 500 µg/mL; Hexal AG, Holzkirchen, Germany) and Dulbecco’s Phosphate-Buffered Saline (DPBS; Sigma-Aldrich GmbH, Steinheim, Germany). Group 2 also received CAP treatment after the development of BLM-induced fibrosis as described in detail below. All mice were maintained under specific pathogen-free and controlled conditions (22 °C, 55% humidity, and 12 h day/night rhythm) and had free access to water and chow. The mice received humane care in compliance with the guidelines outlined in the Guide for the Care and Use of Laboratory Animals (Permit Number: 54-2532.1-16/11).

The back of each mouse was shaved (2 cm × 2 cm) at the beginning of the experiment and once per week in the course of the experiment. All animals were subjected to a well-established BLM-induced scleroderma model [73,74,75] based on the study by Yamamoto et al. on BALB/C mice [75] with small modifications adapted for the 129Sv/Ev mouse strain. The mice received local subcutaneous injections of 100 µL BLM 5 times a week within 6 weeks as well as further BLM injections 5 times per week over the 10 days of CAP treatment to maintain the local sclerosis status. Group 1, which was not treated with CAP, received BLM injections for another 10 days. To induce a comparable level of local fibrosis that was as homogeneous as possible, injections were always given by the same person at the same injection site on the murine back. A further area on the back of each mouse was treated with a subcutaneous injection of 100 µL of DPBS instead of BLM and used as control. Group 1 (8 female and 8 male animals) received no CAP treatment, whereas group 2 (8 female and 8 male animals) received CAP therapy for 10 days (5 times per week; 2 min). The mice were treated with the MicroPlaSterβ^®^ plasma torch system with a distance of 2 cm to the electrode of the plasma device according to the treatment modality established for humans [47] and in other animal studies [59,60]. To achieve comparable stress parameters in all animal groups, the untreated animals (group 1) were placed under the CAP electrode in an analogous manner, but without ignition of any plasma. The behavior of all animals (group 1/group 2) during/after treatment was documented in the animal experiment evaluation form (Permit Number: 54-2532.1-16/11) and showed no differences between the two groups.

After the 10-day CAP therapy, all animals were killed by cervical dislocation, and the BLM-treated and DPBS-treated skin area was removed and subdivided for histological preparation or molecular biological examination.

### 2.4. Isolation of Ribonucleic Acid (RNA) and Reverse Transcription

RNA from hNF, hAF and hLSF was isolated 6 and 24 h after CAP treatment using the NucleoSpin^®^ RNA Kit (Macherey-Nagel, Düren, Germany) according to the manufacturer’s instructions. cDNA was generated with the AMV reverse transcriptase kit (Promega, Mannheim, Germany) using 2–5 µg of total RNA for transcription.

### 2.5. Quantitative Real-Time Polymerase Chain Reaction (PCR) Analysis

Gene expression analysis consisted of quantitative real-time PCR with specific sets of primers (Sigma Aldrich, Steinheim, Germany) and conditions (Table 1) and was performed using LightCycler technology (Roche Diagnostics, Mannheim, Germany) as described elsewhere [67]. PCR reactions were evaluated by melting curve analysis. Beta-actin (β-actin) was amplified to ensure cDNA integrity and to normalize expression. Each experiment was repeated at least three times in duplicates.

### 2.6. Immunohistochemical Analysis

Immunohistochemistry was conducted on formalin-fixed and paraffin-embedded full-skin preparations from control skin treated with Dulbecco’s Phosphate-Buffered Saline (DPBS) (*n* = 16) and from BLM-induced sclerosis skin (*n* = 16) ± CAP treatment (total *n* = 64 full-skin preparations). Skin sections measuring 2 μm were stained with hematoxylin and eosin (H&E) and Sirius Red/Fast Green as described previously [76]. Sirius Red staining intensity from the tissue sections was quantified with the image J software (http://www.imagej.softonic.de; accessed on 21 December 2020) and expressed as percentage of the total area as described elsewhere [72]. Immunohistological staining directed against CD68 (KP1; Dako, Carpinteria, CA, USA) and αSMA (EPR5368; Abcam, Berlin, Germany) was performed according to the method described previously [27,77]. The staining results were evaluated semiquantitatively by means of light microscopy. CD68- and αSMA-positive cells were counted in three non-overlapping high power fields per full-skin preparation (hpf; magnification 40×). All representative images were performed with 20× magnification.

### 2.7. Immunofluorescence Analysis

For F-actin immunofluorescence analysis, 10,000 hNF, hAF and hLSF each were grown on Falcon^®^ 4-well Culture Slides (Corning GmbH; Kaiserslautern, Germany) for 24 h. Afterwards, cells were treated with CAP for 2 min or remained untreated and were further incubated under cell culture conditions for 24 h as described in 2.2. For F-actin visualization, the F-actin Visualization Biochem Kit from Cytoskeleton, Inc. (Denver, CO 80223, USA) was used according to the manual instruction. Images were collected by fluorescence microscopy (Axio Imager Z1; Carl Zeiss Vision GmbH, Halbergmoos, Germany) with 40× magnification.

### 2.8. Spheroid Migration Assay

“Hanging drop” spheroids were generated with hNF, hAF and hLSF according to a modified protocol published by Ruedel et al. [78]. Cells were trypsinized, adjusted to 50,000 cells/mL in DMEM, and mixed with 20% methocel (6 g methyl cellulose; Sigma-Aldrich, Munich, Germany, 250 mL DMEM). A total of 25 mL of the cell suspension were dropped onto the cover of a 9 cm petri dish (Greiner Bio One; Frickenhausen, Germany) filled with DPBS. The cover dish was inverted and incubated under a humidified atmosphere of 8% CO_2_ at 37 °C for 72 h. “Hanging drop” spheroids were carefully inverted and treated with CAP for 2 min in the petri dish or remained untreated. Subsequently, spheroids were harvested and collected in a 15 mL Falcon™ tube (Fisher Scientific GmbH; Schwerte, Germany) filled with DMEM. Harvested spheroids were embedded into a collagen matrix (1 part × 10 minimum essential medium, 1 part 7.5% sodium bicarbonate solution; Sigma-Aldrich, Munich, Germany), and rat tail Coll I (BD Biosciences, Heidelberg, Germany) at a final concentration of 2.5 mg/mL and covered with 2 mL of DMEM and 1.6 mg/mL of mitomycin D (Sigma-Aldrich, Munich, Germany) per well. Migration of fibroblasts out of the spheroid was photographed, and the migrated area was determined immediately (0 h) as well as 24 and 48 h after the start of the assay.

### 2.9. Co-Culture Assay

Co-culture assays with human epidermal keratinocytes (hEK) and fibroblasts (hNF, hAK and hLSF) were performed to study the paracrine effects of keratinocytes on fibroblasts after CAP treatment. This model resembles human skin as it allows the concomitant cultivation of keratinocytes and fibroblasts, just separated by a membrane. The soluble factors produced by keratinocytes after direct CAP treatment are able to permeate this membrane, thus influencing the underlying fibroblasts. For this co-culture assay, 10^5^ hEK were seeded into the upper chamber (Falcon^®^ Permeable Support for 6-well Plate with 0.4 µm Transparent PET Membrane; Corning GmbH; Kaiserslautern, Germany) until they were confluent (after 5–7 days). Confluent hEK were then CAP treated for 2 min or remained untreated in the transwell and subsequently placed above 10^6^ fibroblasts (hNF, hAK and hLSF) cultivated in a 6-well plate (Costar^®^ Multiple Well Cell Culture Plates; Corning GmbH; Kaiserslautern, Germany). Keratinocytes and fibroblasts remained there under cell culture conditions for 6–24 h (2.2.). After 6 and 24 h, fibroblasts were taken for RNA isolation and reverse transcription (2.4.).

### 2.10. Statistical Analysis

All data were analyzed with GraphPad Prism 9 software (GraphPad Software Inc., San Diego, CA, USA) and expressed as mean/median ± standard deviation (SD). The Student’s unpaired *t* test with the nonparametric Mann–Whitney test was used to compare the medians between two groups. One-way ANOVA with Dunnett’s multiple comparison test was used to compare the mean of hAF and hLSF to the mean of hNF. Ordinary one-way ANOVA with Tukey’s multiple comparison test was done to indicate differences of the mean within the untreated or the CAP-treated animal groups. Significant results are indicated * *p* < 0.05, ** *p* < 0.01, *** *p* < 0.001, **** *p* < 0.0001, ^#^ *p* < 0.05, ^##^ *p* < 0.01, ^###^ *p* < 0.001, ^####^ *p* < 0.0001, ns = not significant. A detailed description of the used statistics can be found in the corresponding legend below the respective figure.

## 3. Results

This study investigated the effect of CAP treatment on localized scleroderma (LS) by means of different cell culture models (mono-culture, co-culture) with different fibroblasts (hNF, hAF, hLSF) and an in vivo murine model of bleomycin (BLM)-induced fibrosis. Human normal fibroblasts (hNF) isolated from the dermis of healthy donors were treated with TGF-β to obtain an activated myofibroblast-like phenotype (hAF). hLSF were isolated from the dermis of patients with localized scleroderma (LS). hAF and hLSF were used as in vitro models for the investigation of LS and were compared with hNF.

### 3.1. Pro-Fibrotic Gene Expression in Untreated and CAP-Treated hNF, hAF and hLSF

To examine the expression of pro-fibrotic genes after CAP treatment on various fibroblasts, real-time expression analyses of genes known to be regulated by CAP were carried out 6 and 24 h after CAP treatment. Coll I is known to be induced in LS-derived fibroblasts and was shown to be inducible in hNF after CAP treatment in an earlier study on wound healing [60]. The present study confirmed these results in hNF 6 and 24 h after CAP treatment (Figure 1a,b). On the other hand, Coll I expression in CAP-treated hAF and hLSF did not significantly change after CAP treatment compared to the corresponding untreated control (Figure 1a,b). Similar expression patterns were observed for alpha smooth muscle actin (αSMA) (Figure 1c,d) and fibroblast activating protein (FAP) (Figure 1e,f). These results suggest that CAP can activate pro-fibrotic genes in hNF. In contrast, CAP does not affect the expression of these genes in the two in vitro model systems for LS, in which the basal expression of pro-fibrotic markers is already strongly induced.

### 3.2. Migration of CAP-Treated hNF, hAF and hLSF

A further hallmark of fibrotic processes is the elevated migratory ability primarily of fibroblasts. Migration after CAP treatment was analyzed in a 3-dimensional (3-D) spheroid model. Several studies have shown that CAP promotes cell migration [56,79,80]. In our previous analyses of 2-dimensional (2-D) cell cultures with normal fibroblasts, migration was induced after 30 s of CAP treatment with the MicroPlaSterβ^®^ device [60]. Treatment for 2 min, however, did not improve migration [60]. We hypothesized that this lack of improvement is due to the known attachment loss of cells after high-dose CAP treatment and a delay in starting migration caused by reattachment. Using this 3-D spheroid model, the reattachment process can be bypassed, enabling the direct analysis of the migration effects after 2 min CAP treatment. A total of 24 and 48 h after 2 min CAP treatment, migration of hNF out of the 3-D spheroids was increased (Figure 2a), whereas migration of hAF (Figure 2b) and hLSF (Figure 2c) was significantly reduced in comparison to the corresponding control. Figure 2d gives an exemplary overview of hNF, hAF and hLSF migration out of spheroids immediately (0 h) as well as 24 and 48 h after 2 min CAP treatment or without CAP treatment (ctrl.). In summary, these results indicate that CAP treatment reduces migration in hAF and hLSF and may thus contribute to the anti-fibrotic effect of CAP within these cells.

### 3.3. Expression of Key Mediators in the Induction of Fibrogenesis in Scleroderma Analyzed in hNF, hAF and hLSF Co-Cultured with CAP-Treated hEK

To test whether plasma affects key mediators in the induction of fibrosis [81], the expression of IL-6, TNFα, MMP-1, Coll I and αSMA was examined in fibroblasts co-cultured with CAP-treated hEK for 2 min. Cytokine mRNA expression of IL-6 and TNFα was analyzed 6 h after CAP treatment, whereas MMP-1, Coll I and αSMA in hNF, hAF and hLSF was determined after 24 h. Interestingly, we could show that CAP-treated hEK were able to regulate these genes in fibroblasts via paracrine mechanisms; whether CAP-treated hEK induce or repress these genes in fibroblasts, however, strongly depends on the fibroblast phenotype. The expression of IL-6 and TNFα was significantly induced in hNF, not affected in hAF, and strongly downregulated in hLSF co-cultured with CAP-treated hEK in comparison to the corresponding control (Figure 3a,b). MMP-1 is known to be negatively regulated by TGF-β [11] and was accordingly reduced in TGF-β- stimulated hAF in comparison to hNF (Figure 3c). However, hAF and hLSF co-cultured with CAP- treated hEK significantly induced MMP-1 in comparison to the corresponding untreated control (Figure 3c). The pro-fibrotic marker Coll I and αSMA were induced in hNF co-cultured with CAP-treated hEK (Figure 3d,e), similar to the results after direct CAP treatment (Figure 1b,d). In hAF and hLSF, however, expression was not significantly affected after co-culture with CAP-treated hEK (Figure 3d,e). These results suggest that CAP is able to regulate scleroderma-associated genes via paracrine mechanisms, depending on the fibroblast phenotype and probably due to the basal expression of these genes within the cells.

### 3.4. Examination of the Cytoskeletal Organization in hNF, hAF, and hLSF after CAP Treatment

Myofibroblast differentiation plays a critical role in the pathogenesis of fibrosis [82]. In comparison to the spindle-like shape of hNF (Figure 4a), treatment with recombinant TGF-β in the hAF model system promotes stress fiber (F-actin) formation and myofibroblast differentiation (Figure 4b). In hLSF, elevated endogenous TGF-β production and secretion resulted in a myofibroblast-like phenotype (Figure 4c). CAP treatment for 2 min resulted in the modified organization of the actin cytoskeleton determined by F-actin immunofluorescence staining in hAF (Figure 4e) and hLSF (Figure 4f). Stress fibers were reduced and increasingly transferred to the cell periphery (Figure 4e,f), but this modification was not observed in CAP-treated hNF (Figure 4d). These results may lead to the assumption that CAP treatment counteracts the differentiation step into myofibroblast, which is supposed to be of major importance in all forms of fibrosis [83].

### 3.5. Examination of the Dermal Diameter, the Collagen Content and the Presence of Macrophages in the BLM-Induced Fibrosis Model after CAP-Treatment

To get further insight into the impact of CAP treatment on scleroderma, we applied a well-established murine model of sclerotic skin induced by repeated local injection of BLM [75,84,85]. For this purpose, the following tissue samples were taken: control skin (DPBS) ± CAP treatment and fibrotic skin (BLM) ± CAP treatment. Figure 5 shows the procedure of CAP treatment on mice and provides an overview of the corresponding areas of tissue sampling.

Without BLM treatment (DPBS; ctrl. skin), histological analysis did not show any phenotypical differences between the untreated group (1) and the CAP-treated group (2) analyzed by H&E staining. Skin samples with BLM-induced fibrosis differed significantly between the untreated group 1 and the CAP-treated group 2 (Figure 6a). Mean dermal thickness of the control skin without CAP treatment was 227.00 µm (SD ± 17.38) and 251.60 µm (SD ± 10.40) with CAP treatment; mean dermal thickness of the fibrotic skin without CAP treatment was 581.90 µm (SD ± 20.31) and 386.60 μm (SD ± 16.04) with CAP treatment (Figure 7a). This significant reduction in dermal thickness in CAP-treated BLM-induced sclerosis correlates with the diminished collagen deposition in the affected skin after CAP treatment. The collagen content was analyzed histologically using Sirius Red/Fast Green staining (Figure 6b) and image analysis (Figure 7b). Finally, the number of CD68-positive macrophages and αSMA-positive myofibroblasts was determined. Macrophages and myofibroblasts were significantly reduced after CAP treatment in group 2 in comparison to the fibrotic skin samples of group 1 (Figure 6c,d and Figure 7c,d). In summary, these in vivo results indicate that CAP treatment reduces the pro-inflammatory/pro-fibrotic phenotype in BLM-induced fibrosis and has no significant impact on normal skin.

## 4. Discussion

The aim of this study was to evaluate Cold Atmospheric Plasma (CAP) effects on localized scleroderma (LS) using different in vitro and in vivo model systems and to define whether CAP may be considered for the treatment of LS in future clinical trials. Table 2 provides an overview of the significant in vitro (Table 2a) and in vivo (Table 2b) results within this study.

Our results clearly show molecular and cellular differences in the response of isolated fibroblasts to CAP treatment. hNF, which per se express a relatively low level of pro-fibrotic genes, show a significant induction of Coll I, αSMA, and FAP expression after CAP treatment. Similar observations have been published in other studies on wound healing [60,86,87,88]. The induction of diverse cytokines and growth factors after CAP treatment as observed in those studies, for instance IL-1α, IL-1 β, IL-6, IL-8, TNFα, MCP-1 and TGF-β, contributes to accelerated wound healing. In addition, studies on wound healing have also shown the elevated expression of pro-fibrotic molecules such as Coll I and αSMA [60,89,90], which further improve wound healing. Continuous over-expression of these molecules, however, can also have a negative effect and may induce fibrosis [91].

In this study, we could clearly show that fibrosis-associated molecules such as Coll I, αSMA and FAP, which are particularly expressed in both TGF-β-activated hAF and hLSF, are not further inducible after CAP treatment. We assume that the already elevated threshold of these factors in such cells cannot be further triggered by CAP. Thus, CAP does not seem to be an additional activator of these pro-fibrotic genes in this context. Yet, the genes in these cells were not actively inhibited by CAP at the molecular level. Future studies should focus on the signaling pathways affected by CAP treatment and the causes of the anti-fibrotic effect.

Interestingly, co-culture experiments, in which fibroblasts were cultured together with CAP-treated hEK, yielded a significant decrease in IL-6 and TNFα expression in hLSF. These results may indicate that CAP effects are more likely to be transmitted to fibroblasts via paracrine mechanisms, in this case, through keratinocytes. As physical plasma can only capture cells of the epidermis when locally applied to the skin with a plasma penetration depth of roughly 270 μm [92], such mechanisms are essential for achieving responses in deeper skin layers such as the dermis.

Furthermore, we demonstrated that MMP-1, which is strongly reduced in patients with LS [93] and known to be repressed by TGF-β [11,94], can be significantly induced in hAF and hLSF co-cultured with CAP-treated hEK. This re-expression of MMP-1 in these cells suggest a reduced fibrotic phenotype, although the exact regulatory mechanisms of this process need to be investigated in more detail in further studies.

In addition, our co-culture analyses suggest that CAP effects on Coll I and αSMA gene expression in hAF and hLSF does not seem to be regulated via paracrine mechanisms (keratinocyte/fibroblast exchange), although a CAP-based increase in these pro-fibrotic factors has been observed in hNF. Again, the already elevated basal expression of these pro-fibrotic genes in hAF and hLSF may prevent further induction. Nevertheless, detailed analyses of this topic are still required.

Next, functional migration assays and examination of the cytoskeletal organization in hAF and hLSF in comparison to hNF provided an insight into the potential fibro-protective properties of CAP. Fibroblast transdifferentiation into myofibroblast plays an important role in mediating the fibrotic response to tissue injury and is thought to be a key pathologic step in the pathogenesis of disorders such as LS [81,95]. On the other hand, in wound healing and tissue repair, this specialized contractile cell type is responsible for wound closure, tissue contraction, and scarring by synthetization of interstitial collagens, fibronectins, and other matrix components to repair damaged tissue [81]. In hNF, CAP treatment significantly increases migration, which is not only a pro-fibrotic hallmark but has also the desired effect of promoting wound healing [80]. Interestingly, in hAF and hLSF, migration was significantly reduced after CAP treatment. Together with the observed decrease in stress fiber formation in these cells, we suspect a regression of this fibrosis-associated myofibroblast phenotype after CAP treatment.

Similar opposing results of CAP on cell migration and collagen production comparing normal fibroblasts (NFs) and fibroblasts isolated from keloid patients (KFs) were observed by Kang et al. [96]. Keloid pathogenesis involves extreme fibroblast activity including perturbed proliferation, migration, and collagen synthesis [97]. The authors could show that CAP treatment of KFs downregulates the epidermal growth factor receptor/signal transducers and activators of the transcription 3 (EGFR/STAT3) pathway, which suppresses TGF-β expression, thereby decreasing cell migration and collagen production. Yet, according to Kang et al., CAP treatment upregulated the EGFR/STAT3 pathway as well as collagen production and increased cell migration in NFs. The authors concluded that CAP can therefore have beneficial therapeutic effects on the pathogenesis of keloid scars and on normal wound healing, which are mediated by the EGFR/STAT3 signaling pathway. Whether these regulatory relationships also apply to the pathogenesis of LS has to be analyzed in follow-up studies.

The most impressive anti-fibrotic CAP effect in the present study was seen in vivo in the BLM-induced fibrosis model. In addition to a significant reduction in the dermal diameter of the affected skin areas, we observed a significant decrease in the collagen content with fewer macrophages and myofibroblasts in the affected tissue samples. To our knowledge, no clinical study has yet confirmed such anti-fibrotic CAP effects in humans. However, positive CAP effects have already been observed in a proof-of-concept study describing CAP as a reliable conservative treatment option for complicated wound healing disturbances. This effect was exemplarily shown in the case of morbidity of a radial forearm free flap donor site with exposed flexor tendons [98].

In summary, our study provides the first evidence that CAP may be a complementary treatment option to existing LS therapies. In extensive experience in the treatment of chronic and acute wounds, CAP treatment is neither toxic nor mutagenic and completely painless.

## 5. Conclusions

CAP is a rapidly growing new research area in health care. One promising novel medical application of CAP is the treatment of fibrotic lesions such as LS. In this study, we could show for the first time that CAP treatment suppresses cell migration of hLSF, counteracts myofibroblast differentiation, and reduces the presence of macrophages at the site of inflammation but increases these mechanisms in hNF in vitro. Cell migration, myofibroblast differentiation, and the presence of large amounts of macrophages is a hallmark of wound healing and is also important for LS formation. Thus, suppression of hLSF and macrophage activation by CAP may be a novel therapeutic strategy for the treatment and prevention of LS.

From the observations made in the animal study, it is not possible to draw direct conclusions about the efficacy of CAP on LS patients. However, the observed anti-fibrotic effects after CAP treatment may be the basis for clinical trials.

## Figures and Tables

**Figure 1 biomedicines-09-01545-f001:**
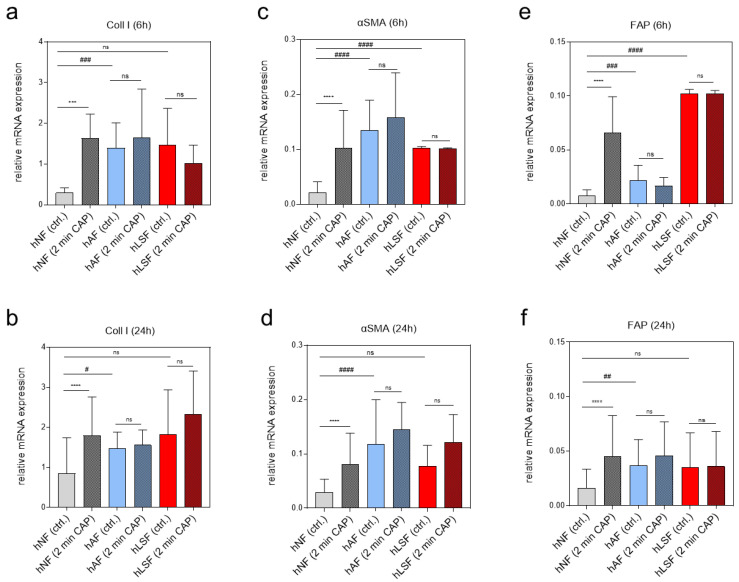
Effects of CAP on pro-fibrotic gene expression in normal, activated and LS-derived fibroblasts. The relative mRNA expression of Coll I (**a**,**b**), αSMA (**c**,**d**) and FAP (**e**,**f**) in hNF, hAF and hLSF was analyzed 6 (**a**,**c**,**e**) and 24 h (**b**,**d**,**f**) after CAP treatment. Statistical analysis: The Student’s unpaired *t* test and the nonparametric Mann–Whitney test were used to show differences between the medians of two groups. Significant results compared to the corresponding untreated control are indicated *** *p* < 0.001, **** *p* < 0.0001; ns = not significant. One-way ANOVA with Dunnett’s multiple comparison test was done to compare the mean of hAF and hLSF to the mean of hNF ^#^ *p* ≤ 0.05, ^##^ *p* < 0.01, ^###^ *p* < 0.001, ^####^ *p* < 0.0001; ns = not significant.

**Figure 2 biomedicines-09-01545-f002:**
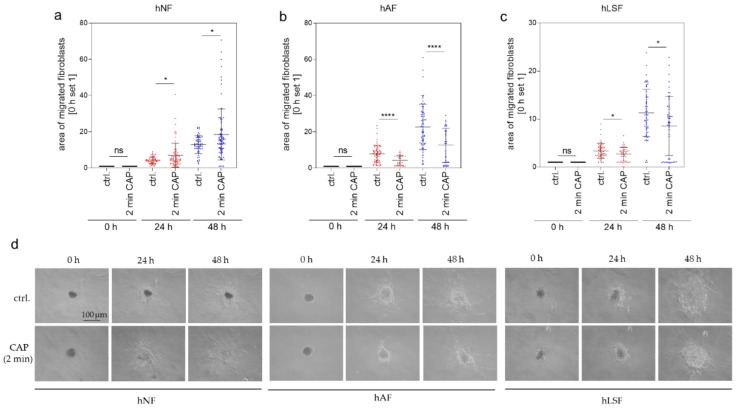
Cell migration out of 3-dimensional (3-D) spheroids. hNF (**a**), hAF (**b**) and hLSF (**c**) spheroids were analyzed for their migratory ability. CAP-treated and untreated spheroids were seeded into a collagen matrix (0 h). The migratory ability of the cells out of the spheroid was analyzed after 24 and 48 h. The spheroid area at time 0 h was set to 1 and was analyzed in comparison to the area determined after 24 and 48 h. The area of the migrated fibroblasts as exemplarily shown in (**d**) was calculated from three independent experiments with *n* indicating the number of total analyzed spheroids (hNF: *n* = 58–82; hAF: *n* = 35–70; hLSF: *n* = 50–61). Statistical analysis: The Student’s unpaired *t* test and the nonparametric Mann–Whitney test were used to show differences between the medians of two groups. Significant results compared to the corresponding untreated control are indicated * *p* ≤ 0.05, **** *p* < 0.0001, ns = not significant. Scale bar = 100 μm.

**Figure 3 biomedicines-09-01545-f003:**
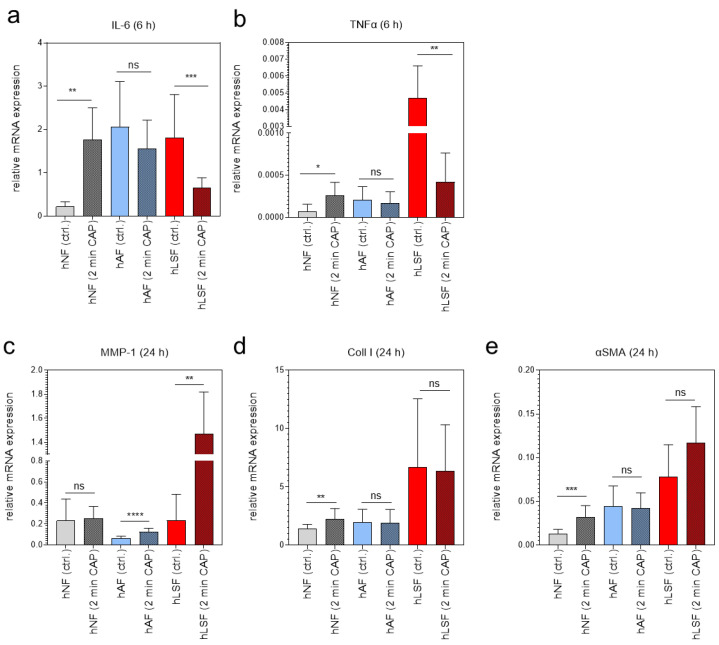
Paracrine effects of CAP on normal, activated, and LS-derived fibroblasts analyzed by mRNA expression of inflammatory- and fibrotic-associated marker genes. The relative mRNA expression of IL-6 (**a**) and TNFα (**b**) was analyzed 6 h, and the expression of MMP-1 (**c**), Coll 1 (**d**) and αSMA (**e**) in hNF, hAF, and hLSF 24 h after co-culture with hEK CAP-treated for 2 min and compared to the corresponding untreated controls (*n* = 3 in duplicated form). Statistical analysis: The Student’s unpaired *t* test and the nonparametric Mann–Whitney test were used to show differences between the medians of two groups. Significant results compared to the corresponding untreated control are indicated * *p* ≤ 0.05, ** *p* < 0.01, *** *p* < 0.001, **** *p* < 0.0001; ns = not significant.

**Figure 4 biomedicines-09-01545-f004:**
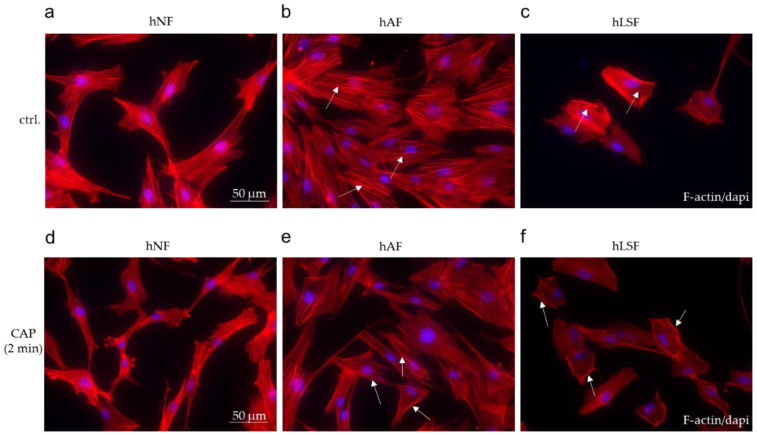
The effect of CAP on the cytoskeletal organization in fibroblasts. Immunofluorescence staining for F-actin/dapi on untreated hNF, hAF and hLSF (**a**–**c**) and after CAP treatment for 2 min (**d**–**f**). The dashed arrows in (**b**,**c**) mark the strong stress fiber formation within the activated fibroblasts. The solid arrows in (**e**,**f**) indicate the pheripheral localization of the stress fibers after CAP treatment. Scale bar = 50 μm.

**Figure 5 biomedicines-09-01545-f005:**
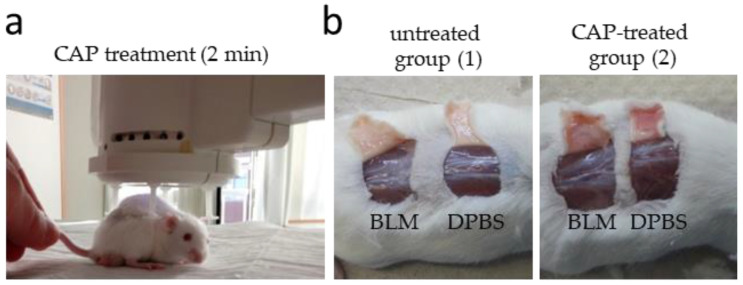
CAP treatment of mice and tissue sampling. Prior to CAP treatment, dermal fibrosis was induced in defined areas on the murine back by 100 µL of BLM injection (500 µg/mL) for 28 days. Corresponding control skin was obtained from the same animal by 100 µL of DPBS injection for 28 days to exclude reactions due to injection into the affected skin. Afterwards, the backs of the mice remained untreated (group 1) or were treated with CAP for 2 min (group 2) using the MicroPlaSterβ^®^ device. In total, the animals received 10 CAP treatments five times per week as shown in (**a**). After 10 day treatment with CAP, all mice, including the untreated ones, were killed, and the corresponding skin areas were removed (**b**).

**Figure 6 biomedicines-09-01545-f006:**
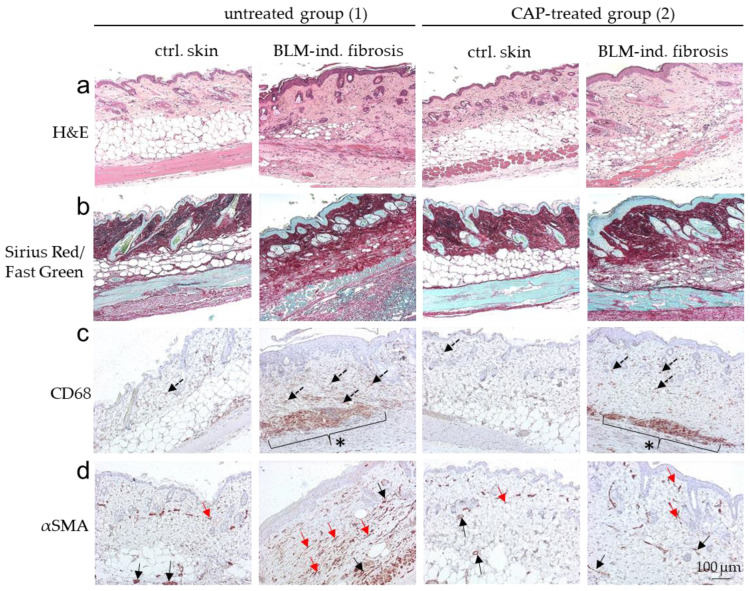
Histological overview of the BLM-induced scleroderma model after CAP treatment. Representative images of (**a**) H&E, (**b**) Sirius Red/Fast Green, (**c**) CD68- and (**d**) αSMA-stained sections of untreated (group 1) and CAP-treated (group 2) mice. Each group contains skin sections treated with DPBS (ctrl. skin) and bleomycin (BLM-ind. fibrosis) collected from two separate areas on the same murine back. Dotted arrows in (**c**) indicate CD68-positively stained macrophages elevated in BLM-induced fibrosis and a reduction in dermal tissue after CAP treatment. * in (**c**) indicates an increase in capillary macrophages in BLM-induced fibrosis in comparison to control skin. This increase of vascular macrophages was not included in the evaluation of CD68 staining. αSMA-stained myofibroblasts (red arrows) in (**d**) are strongly induced in BLM-induced fibrosis. Vessel structures are also αSMA-stained (black arrows) in (**d**) and were excluded in the evaluation. Scale bar: 100 µm.

**Figure 7 biomedicines-09-01545-f007:**
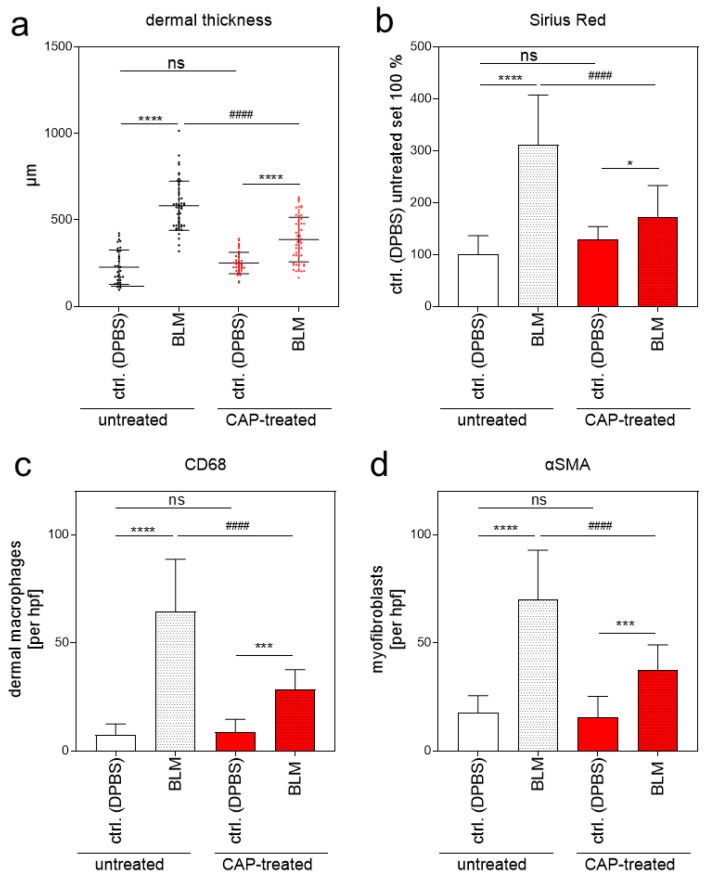
Evaluation of dermal thickness, collagen content, and the number of CD68-positive macrophages and α-SMA-positive myofibroblasts in a BLM-induced scleroderma model after 2 min CAP treatment. (**a**) Dermal thickness was determined in μm in control skin (DPBS) ± CAP treatment, and in fibrotic skin (BLM) ± CAP treatment. Representative images of the mice (*n* = 32; per group and treatment condition) were taken, and five measurements per image were used for calculation. (**b**) The six representative Sirius Red/Fast Green images per group and treatment condition were used for determining the fibrotic region by quantifying the Sirius Red-positive area by means of image analysis. (**c**) The number of CD68-positive macrophages was determined by counting the positively stained cells (average of three different hpf/animal; *n* = 6 animals per group and treatment condition) using 20× magnification. (**d**) Quantification of αSMA-positive myofibroblasts (average of three different hpf/animal; *n* = 6 animals per group and treatment condition). Statistical analysis: Ordinary one-way ANOVA with Tukey´s multiple comparison test was used with * *p* ≤ 0.05, *** *p* < 0.001, **** *p* < 0.0001 to indicate the mean differences within the untreated or the CAP-treated group. ^####^ *p* < 0.0001 indicates significant changes between untreated and CAP-treated groups in the BLM-induced fibrosis samples; ns = not significant. αSMA, alpha smooth muscle actin; hpf, high-power field.

**Table 1 biomedicines-09-01545-t001:** Human primers and conditions.

Primer Name	Forward Primer 5′→ 3′	Reverse Primer 5′→ 3	Condition ^1^ (Annealing, Melting)
β-actin	CTACGTCGCCCTGGACTTCGAGC	GATGGAGCCGCCGATCCACACGG	ann. 60 °C, melt. 85 °C
Coll I	CGGCTCCTGCTCCTCTT	GGGGCAGTTCTTGGTCTC	ann. 60 °C, melt. 86 °C
αSMA	GGCCGAGATCTCACTGACTAC	TTCATGGATGCCAGCAGA	ann. 58 °C, melt. 84 °C
FAP	CGGCCCAGGCATCCCCATTT	CACTCTGACTGCAGGGACCACC	ann. 60 °C, melt. 76°C
IL-6	GGTACATCCTCGACGGCATCT	GTGCCTCTTTGCTGCTTTCAC	ann. 60 °C, melt. 79 °C
TNFα	ATCCTGGGGGACCCAATCTA	AAAAGAAGGCACAGAGGCCA	ann. 60 °C, melt. 81 °C
MMP-1	TCACCAAGGTCTCTGAGGGTCAAGC	GGATGCCATCAATGTCATCCTGAGC	ann. 65 °C, melt. 80 °C

^1^ Quantitative real-time PCR was conducted with specific sets of primers and conditions. ann: annealing temperature; melt: melting temperature.

**Table 2 biomedicines-09-01545-t002:** Overview of CAP treatment effects on hNF, hAF, hLSF and BLM-induced fibrosis.

(**a**) **CAP Treatment Effects In Vitro**	**hNF**		**hAF**		**hLSF**	
	ctrl.	CAP	ctrl.	CAP	ctrl.	CAP
pro-fibrotic gene expression (Coll I, αSMA, FAP) in mono-culture	↑	↑↑	↑↑	↔	↑↑	↔
pro-inflammatory gene expression (IL-6, TNFα) in co-culture	↑	↑↑	↑↑	↔	↑↑	↓
pro-fibrotic gene expression (Coll I, αSMA) in co-culture	↑	↑↑	↑↑	↔	↑↑	↔
expression of MMP-1 in co-culture	↑↑	↔	↑	↑↑	↑	↑↑↑
migration (48 h after CAP treatment)	↑	↑↑	↑↑	↓↓	↑↑	↓↓
cytoskeletal organization (F-actin fibers)	↑	↔	↑↑	↓↓	↑↑	↓↓
(**b**) **CAP Treatment Effects In Vivo**	**Untreated Group (1)**			**CAP-Treated Group (2)**		
	ctrl. skin	BLM-ind. fibrosis		ctrl. skin	BLM-ind. fibrosis	
dermal thickness	↑	↑↑↑		↑	↑↑	
collagen content	↑	↑↑↑		↑	↑↑	
macrophages	↑	↑↑↑		↑	↑↑	
myofibroblasts	↑	↑↑↑		↑	↑↑	

Note: ↑/↓ indicates moderate induction/reduction, ↑↑/↓↓ indicates significant induction/reduction, ↑↑↑ indicates strong significant induction/reduction, ↔ indicates no change compared to untreated ctrl.; BLM: bleomycin; hNF: human normal fibroblasts; hAF: human TGF-β-activated fibroblasts; hLSF: human localized scleroderma-derived fibroblasts; Coll I: collagen type I; αSMA: alpha smooth muscle actin; FAP: fibroblast activating protein; IL-6: interleukin 6; TNFα: tumor necrosis factor alpha; MMP-1: matrix metalloproteinase 1.

## Data Availability

The data used to support the findings of this study are included in the article. The data sets generated during and/or analyzed during the current study are available from the corresponding author on request.

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
