# Peer review of "The Anti-Fibrotic Effect of Cold Atmospheric Plasma on Localized Scleroderma In Vitro and In Vivo"

_biomedicines, 2021, doi:10.3390/biomedicines9111545_

Round 1

Reviewer 1 Report

The presented research is very interesting. The research is well designed and carried out. Introduction contains enough background informations.

Materials and methods are clearly described. The statics are not carried out correctly. One wonders why when using the nonparametric Mann-Whitney test the results are shown as mean and not median?

References are presented carelessly. Once, the authors provide abbreviations of journals and once full names, e.g. items 1 and 3.

Tables presented in the discussion section should be included in the results section.

Reviewer 2 Report

The article is very complete, just an observation, it would be very convenient to indicate that the applied energy does not cause any damage to the tissue (although they show it)

Reviewer 3 Report

Dear Authors,

although your study gives some hints that CAP may represent a potential treatment for scleroderma, in particular, the methods and study design are not sufficient.

First of all and this is the main problems of the majority of CAP studies:

The lack of controls!

Animal experiments:

You work with argon gas flow of 4 l/min, which in turn has effects on skin (cooling, oxygen displacement, drying effects). Furthermore, CAP treatments are stressfuls for animals, which also effects healing, inflammation and other biological processes.

Thus, you need a least a gas control treatments (same argon flow and temperatur) and a shame treatment without argon/CAP to observe stress-induced effects by treatment. To study the impact of stress by the procedure of CAP treatment another control group without any intervention exept BLM treatment is necessary.

It is not even clear that you  performed a shame treatment with argon (flow/temperatur).

Cell experiments:

You described in your paper

Cold Atmospheric Plasma (CAP) Changes Gene Expression of Key Molecules of the Wound Healing Machinery and Improves Wound Healing In Vitro and In Vivo

that CAP 2 min led to deattchment of cells.

How you can be sure that in your present study CAP did not induce cell death, deattachment or other effects? For example, myofibroblasts attach stronger than fibroblast, thus, by CAP your performed a selection in favor of myofibroblast.

Also, an argon gas flow of 4 l/min led to CAP unspecific effects such excessive drying, hypoxia and hyperosmolarity, especially, when you treated a small volume 750 µl in 35 mm dishes. In my experience, gas flow >500 ml/min can induce signficiant toxicity under these conditions. Thus, you need here also a gas flow controls and untreated controls.

Furthermore, you do not give some explanations or propose some mechanism who CAP may reduce fibrosis. How it enters through epidermis and has effects of BLM treated mice, whereas in DPBS no effects could be observed.

Finally, hairy shaved mouse skin is very different from human skin. This is a limitation of your study you forgot to discuss.

U

In my experience

Round 2

Reviewer 3 Report

Comments:

You added that:

This device has been successfully used for the treatment of chronic and acute wounds [47,68,69], skin pruritus [42], actinic keratosis [70] in both clinical studies and in different in vitro and in vivo animal models [59,60]. In addition, the evidence that neither toxic nor mutagenic effects on cells could be observed after treatment with the MicroPlaSterβ® device has been published [71]. Tissue or cell damage using the MicroPlaSterβ® or argon gas placebo mode was not reported in any of these studies, which is why we did not perform argon gas controls in this study.

Unfortunately, these mentioned studies are not masterpieces in respective to study design and necessary controls.

Study [47]: No gas control, the reduction of bacterial load is not even a 1 log10 decrease, thus clinical not relevant

Study [68]: Here, a gas control but not a non-treatment control. Do you not know that a drying of wounds with argon or CAP may influence the outcome?

Study [69]: Again, no gas control and reduction of bacterial load too low to be clinical relevant.

Study [42]: Here, only gas controls but not non-treatment controls.

Results did not show any difference between gas control and CAP treatment against pruritus. However, patients reported improvement for both- Thus, argon alone can represent a treatment option?

Study [70]: It is only small study published as a letter to the editor without any controls! Still waiting for a clinical trial…

Study [59] and [60]: It is not clear if there is gas control or what placebo mode means. Nevertheless, 2 groups is not enough. Thus, there are no gas controls or there are no non-treatment control.

Study [71]: You showed an increase in toxicity and a doubling! of mutagenicity by CAP treatment.

By performing only 3 experiments followed by one way ANOVA multi for analysis it is not a surprise that these results did not achieved statistical significance. But to claim that there are no damaging effects at all is a pretty bold assumption.

That these studies with so many flaws could be published somehow is in my opinion very critical,

unfortunately also very common in almost all studies dealing with cold plasma.

But this should not be reason to not perform necessary controls in the submitted study.
